# Methods Used to Evaluate the Immediate Effects of Airway Clearance Techniques in Adults with Cystic Fibrosis: A Systematic Review and Meta-Analysis

**DOI:** 10.3390/jcm10225280

**Published:** 2021-11-13

**Authors:** Naomi Chapman, Kathryn Watson, Tamara Hatton, Vinicius Cavalheri, Jamie Wood, Daniel F. Gucciardi, Elizabeth F. Smith, Kylie Hill

**Affiliations:** 1Curtin School of Allied Health, Faculty of Health Sciences, Building 401 Curtin University, Kent Street, Bentley, WA 6102, Australia; naomi.chapman2@health.wa.gov.au (N.C.); d.gucciardi@curtin.edu.au (D.F.G.); k.hill@curtin.edu.au (K.H.); 2Physiotherapy Department, Sir Charles Gairdner Hospital, Hospital Ave., Perth, WA 6009, Australia; kathryn.watson@health.wa.gov.au (K.W.); tamara.hatton@health.wa.gov.au (T.H.); 3Institute for Respiratory Health, Perth, WA 6009, Australia; 4Curtin enAble Institute, Faculty of Health Sciences, Building 408 Curtin University, Kent Street, Bentley, WA 6102, Australia; 5Allied Health, South Metropolitan Health Service, Perth, WA 6150, Australia; 6Abilities Research Center, Department of Rehabilitation and Human Performance, Icahn School of Medicine at Mount Sinai, New York City, NY 10029, USA; jamie.wood@mountsinai.org; 7Children’s Lung Health Team, Wal-Yan Respiratory Research Centre, Telethon Kids Institute, Perth Children’s Hospital, 15 Hospital Ave., Perth, WA 6009, Australia; elizabeth.smith@telethonkids.org.au

**Keywords:** cystic fibrosis, adults, airway clearance, chest physiotherapy, outcome measures

## Abstract

This review reports on methods used to evaluate airway clearance techniques (ACT) in adults with CF and examined data for evidence of any effect. Sixty-eight studies described ACT in adequate detail and were included in this review. Frequently reported outcomes were sputum expectoration (72%) and spirometric lung function (60%). Compared with cough alone, following any ACT, there was a trend for greater sputum wet weight, however FEV_1_ was not different. The mean (95% CI) within-group effect for sputum wet weight following any ACT was 12.43 g (9.28 to 15.58) (*n* = 30 studies) and for FEV_1_ was 0.03 L (−0.17 to 0.24) (*n* = 14 studies). Meta-regression demonstrated that, when compared with cough alone, greater sputum wet weight was reported in groups that received additional ACT by between 2.45 and 3.94 g (F_3,66_ = 2.97, *p* = 0.04). These data suggest the addition of ACT to cough alone may optimise sputum clearance; however, FEV_1_ lacked sensitivity to detect this change. Importantly, this review highlights the lack of appropriate measures to assess ACT efficacy.

## 1. Introduction

Cystic fibrosis (CF) is a life-limiting inherited disease, with progressive respiratory impairment being the leading cause of morbidity and mortality [1,2,3]. Lung disease in CF is caused by excessive viscous airway secretions and chronic infection, which reduces mucociliary clearance leading to inflammation and destruction of airway walls [4]. Airway clearance techniques (ACT) are an important component of physiotherapy care for people with CF, facilitating mucociliary clearance and reducing sputum load in the lungs with the overall goal of reducing exacerbation frequency and slowing disease progression [5,6]. Airway clearance techniques are defined as “any technique which manipulates lung volumes, gas flow, ventilation, gravity, pulmonary pressures and/or compressive forces with the goal of shearing sputum along the airway lumen towards the mouth” [7]. Commonly used ACT can be grouped according to their mechanism of action, and classified as those which (i) deliver flow to create positive pressure within the airways (positive-pressure device (PPD); e.g., non-invasive ventilation [NIV]); (ii) use a device to create positive intrinsic pressure within the airways during expiration with or without oscillation (positive expiratory pressure device (PEP); e.g., PARI PEP, Aerobika, Acapella, Flutter); (iii) do not utilise positive pressure but require the person to manipulate their breathing pattern to change flow and/or volume beyond the third generation airways, which may be applied with or without external vibration/percussion of the chest wall (other techniques (OT)); or (iv) involve spontaneous/directed coughs performed in isolation from other ACT (cough alone (C)) [8]. Several objective outcomes have been used to evaluate the immediate effects of ACT as well as the medium- and long- term effects of these techniques. However, there is little agreement about which are to be used at each time interval post ACT. The primary outcome for respiratory function recommended by the European Medicines Agency [9] and the US Food and Drugs Administration [10] for clinical trials in people with CF is spirometry, in particular, forced expiratory volume in one second (FEV_1_). Other examples of objective outcomes used to evaluate the immediate effects of ACT include sputum expectoration (as either wet/dry weight or volume), inflammatory markers and rheology, measurements of lung volumes, lung mechanics, diffusion capacity and various imaging modalities. Despite the common use of ACT in this clinical population, no study has attempted to comprehensively synthesise the methods used to evaluate the immediate effects of ACT in adults with CF.

Therefore, the primary aim of this review was to synthesise data from studies that had applied ACT in adults with CF to determine (i) which ACT have been used in this clinical population and (ii) what outcomes have been used to evaluate the immediate effect of ACT. An exploratory aim was to determine in this same population whether there is any evidence that ACT, when compared to cough alone, results in a measurable change, using commonly used outcomes.

## 2. Materials and Methods

The review has been reported in accordance with the Preferred Reporting Items for Systematic Reviews and Meta-Analyses (PRISMA) guidelines [11]. A protocol was not published; however, it was registered with PROSPERO (CRD42020155843).

### 2.1. Study Criteria

To be included in this review, studies (of any design excluding reviews) needed to have applied ACT to adults (mean age ≥ 18 years) with CF and collected data on outcomes to evaluate the effect of ACT within 60 minutes of treatment completion. Neither nebulised mucolytics nor exercise alone were considered ACT. Studies were excluded if (i) participants within a group received different ACT and individual data were unable to be obtained (e.g., a group that continued with their usual ACT, which differed between participants); (ii) they were not written in English; or (iii) they were available only as abstracts. 

### 2.2. Search Procedure

Studies were identified by searching the following databases from inception to August 2021: the Cochrane Library, CINAHL, Embase (via OVID), PEDro and PubMed. The search strategy used for PubMed can be found in Appendix B. This was adapted for use in other databases. Reference lists from included studies were also screened.

### 2.3. Screening

Studies identified in the database search were imported into Covidence [12] and screened independently based on their title and abstract by two review authors (NC and KW). Studies that were unable to be excluded based on title and abstract were screened by reading the full text. Any disagreement between the two review authors about inclusion of studies was resolved via discussion with a third review author (KH). 

### 2.4. Data Extraction and Coding

Data were extracted by NC on authors, publication year, participant characteristics, sample size, study setting, and type and duration of any ACT. The ACT performed by each group within each study was coded according to its mechanism of action as positive-pressure device (PPD), positive expiratory pressure device (PEP), other techniques (OT) or cough alone (C). For groups that used an ACT that utilised more than one mechanism of action, the group was coded according to the most ‘active’ mechanism in the hierarchy of: C < OT < PEP < PPD (e.g., a study that applied the active cycle of breathing techniques (an OT) with concurrent non-invasive ventilation (PPD) was coded as PPD). Data were extracted on objective outcomes used to evaluate the effect of the ACT at a single time point closest to (but not extending beyond) 60 minutes following treatment completion. Data were not extracted on outcomes collected to monitor patient tolerance/safety (i.e., heart rate, blood pressure or oxygen saturation) or to report the flow characteristics of an ACT. For papers published after 2005, study authors were contacted in the case of missing, incomplete or ambiguous data. Numerical data were estimated from figures using the software Graph Grabber (Quintessa; 2.0.2, Oxfordshire, UK).

### 2.5. Risk of Bias

For randomised controlled trials (RCT) and randomised cross-over trials (RXT), risk of bias was assessed using the Revised Cochrane Risk of Bias tool for randomised trials (RoB 2) [13]. This tool assesses risk of bias across six domains and rates each domain and the overall study as being at low, high or unclear risk of bias. Data were presented using a data visualisation tool [14]. Risk of bias was assessed for RCT and RXT only as other study designs included are expected to have high or unclear risk of bias.

### 2.6. Data Analysis

For the primary aims, data on the type of ACT and the outcomes used to evaluate the immediate effect of the ACT were summarised as a narrative synthesis. Further information regarding data analyses of the primary outcomes can be found in the Appendix A. 

### 2.7. Exploratory Analysis

Where possible, studies that compared one type of ACT (i.e., PPD, PEP or OT) with C were included in a meta-analysis (RevMan Web, Cochrane; 5.1.2, Copenhagen, Denmark) [15] to derive the pooled between-group mean difference and 95% confidence interval (CI). A random-effects model was used unless there was no statistical heterogeneity (i.e., I^2^ = 0), in which case a fixed-effects model was chosen. Sensitivity analyses were conducted when I^2^ approached or exceeded substantial heterogeneity (i.e., 50%). Given that earlier work in this area has reported that there is significant heterogeneity in the types of ACT and outcomes utilised as well as limited data described in sufficient detail for meta-analysis [5,16,17,18], exploratory analyses were undertaken to determine whether the mechanism of action for the ACT moderated the magnitude of any within-group changes. To do this, where data were reported for the same outcome, in the same units of measurement, across ≥ four studies, pooled estimates of the within-group change were provided via a three-level meta-analysis (R package, Metafor; 4.0.3, Maastricht, The Netherlands) [19]. Meta-regression analysis was performed for sputum wet-weight, dry-weight and FEV_1_, as 10 or more studies reported these outcomes, in the same units of measurement [20], to determine whether the magnitude of within-group change was moderated by the mechanism of action of the ACT (i.e., PPD, PEP, OT or C) or clinical stability of the study sample (i.e., acute exacerbation, stable or mixed/no information). As these analyses included data from RXT, a three-level meta-regression was undertaken to account for dependency among effects within primary studies. For all analyses *p* < 0.05 was used to denote significance.

## 3. Results

### 3.1. Study Characteristics 

A total of 2048 records were identified. The flow of studies has been summarised in Figure 1.

Of the 70 included studies, 3 (4%) were RCT, 57 (81%) were RXT and 10 (14%) were single-group interventional studies (Table 1). Thirty-five studies (50%) were published prior to year 2000. Data were available on 1204 participants (685 (57%) males; (mean ± SD) age ranged between 18 ± 4 and 36 ± 17 years; FEV_1_ ranged between 25 ± 6 and 76% predicted (SD not reported for the latter)).

The assessment of risk of bias for RCT and RXT are presented in Appendix A. Two of the three RCT were judged as having an unclear risk of bias and the third was judged as having a low risk of bias. For the RXT, 6 (11%) were judged as having low risk of bias, 31 (54%) were judged as having a high risk of bias and 20 (35%) were judged as having an unclear risk of bias. All RXT were judged as having a low risk of bias in regard to measurement of the outcome and all but one RXT were judged as having low risk of bias due to missing outcome data. The most common domains for high or unclear risk of bias in RXT were related to bias arising in the selection of the reported results (*n* = 49 studies; 86%), from the randomisation process (*n* = 39 studies; 68%) and from deviations from intended interventions (*n* = 28 studies; 59%).

### 3.2. Coding According to Mechanism of Action 

The ACT were described in adequate detail in 68 (97%) studies (with a total of 150 intervention group’s coded accordingly). Nine (13%) studies utilised PPD, 29 (43%) utilised PEP, 51 (75%) utilised OT and 17 (25%) utilised C. 

### 3.3. Description of Outcomes

Of the 68 studies that had their ACT coded, all reported data on at least one objective outcome within 60 min following completion of the ACT (Figure 2). Those most frequently reported measures were the amount of sputum expectorated, expressed as weight or volume (49 (72%) studies) and spirometry, in particular FEV_1_ (41 (60%) studies).

### 3.4. Magnitude of Between-Group Change

There were sufficient data from RCT and RXT to conduct a meta-analysis to estimate the effect of any ACT on the outcomes of sputum wet weight, dry weight and FEV_1_. These data are presented in the online supplement (Appendix A. There was considerable heterogeneity across studies in regard to measures reported and types of ACT conducted, which limited data that were able to be pooled across studies. The pooled data demonstrated no between-group difference in any of the comparisons for sputum wet weight, dry weight or FEV_1_ (*p* > 0.05 for all). 

### 3.5. Magnitude of Within-Group Change 

#### 3.5.1. Sputum Expectoration

Of the 49 studies that reported the amount of sputum expectorated following ACT, 32 (65%) had data that could be used in the meta-analyses. The pooled estimate (95% CI) for sputum wet weight, dry weight and volume were 12.43 g (9.28 to 15.58) (70 groups, 30 studies, *n* = 487; Figure 3a), 0.42 g (0.19 to 0.66) (28 groups, 11 studies, *n* = 201; Figure 3b), and 8.01 mL (2.18 to 13.84) (9 groups, 4 studies, *n* = 79; Figure 3c), respectively.

Meta-regression revealed that the effect of ACT on sputum wet weight was moderated by the mechanism of action of ACT (F_3,66_ = 2.97, *p* = 0.04) such that those ACT that were grouped as PPD, PEP and OT produced a higher amount of wet weight sputum compared with C alone by 2.45 to 3.94 g; mean (95% CI) PPD 12.75 g (8.61 to 16.88); PEP 13.76 g (10.16 to 17.36); OT 12.27 g (9.02 to 15.52); C 9.82 g (6.00 to 13.65). The estimate of the effect of ACT on sputum wet weight was not moderated by the clinical stability of the study participants (F_3,66_ = 0.11, *p* = 0.96). The estimate of the effect of ACT on sputum dry weight was neither moderated by the mechanism of action of the ACT (F_3,24_ = 0.11, *p* = 0.95) nor by the clinical stability of the participants (F_2,25_ = 0.65, *p* = 0.53). Meta-regression could not be undertaken for sputum volume due to inadequate study numbers. 

#### 3.5.2. Spirometric Measures of Lung Function 

Of the 41 studies that reported FEV_1_ before and after ACT, 22 (53%) had data that could be used in the meta-analyses. The pooled estimate (95% CI) for change in FEV_1_ following ACT, expressed in litres, was 0.03 L (−0.17 to 0.24) (30 groups, 14 studies, *n* = 218; Figure 4a) and expressed as percent predicted, was 1.44% (0.06 to 2.83) (19 groups, 11 studies, *n* = 172; Figure 4b). 

Meta-regression revealed that the effect of ACT on FEV_1_, expressed in litres and percent predicted, was neither moderated by the mechanism of action of the ACT (F_3,26_ = 0.01, *p* = 1.00; F_3,15_ = 0.21, *p* = 0.89, respectively) nor by the clinical stability of the study participants (F_4,25_ < 0.01, *p* = 1.00; F_3,62_ = 0.10, *p* = 0.96, respectively).

#### 3.5.3. Static Lung Volumes

Of the six studies that reported residual volume (RV) before and after ACT, four (57%) had data that could be used in the meta-analyses. The pooled estimate (95% CI) for change in RV following ACT, expressed in litres, was −0.14 L (−0.72 to 0.44) (eight groups, four studies, *n* = 67; Figure 5). Meta-regression could not be undertaken for RV due to inadequate study numbers.

#### 3.5.4. Other Outcomes

Of the six studies that reported measures of sputum rheology and one study that reported measures of sputum inflammatory biomarkers before and after ACT, none had data that could be used in the meta-analyses. Thirteen studies reported peak expiratory/cough flow, six reported measures using inert gas washout, three reported measures of respiratory mechanics, two reported measures of pulmonary diffusion capacity, two reported measures of respiratory muscle strength and seven reported measures quantified through lung imaging before and after ACT. None had data that could be used in the meta-analyses. 

## 4. Discussion

This review synthesised data from studies that were conducted in adults with CF and had evaluated the effects of any ACT within 60 minutes of treatment completion. The main findings were that in adults with CF (i) there was large variability in ACT applied and outcomes used to evaluate their effect; (ii) robust data from randomised trials were lacking which precluded useful meta-analyses of between-group differences; (iii) the magnitude of the immediate within-group change of ACT on sputum wet weight was moderated by the mechanism of action of the ACT such that PPD, PEP and OT appeared to generate more sputum than cough alone; and (iv) FEV_1_ and RV do not appear to be responsive to change immediately following ACT.

Studies exploring the immediate effects of ACT in adults with CF were heterogeneous in terms of both the types of ACT and outcomes used to evaluate their effect. This finding corroborates those of previous systematic reviews that have sought to explore the effect of ACT in CF [5,16,17,91,92,93,94], bronchiectasis [95] and chronic obstructive pulmonary disease (COPD) [96]. The dates of publication presented in Table 1 suggest that ACT have evolved over the years, with a move from passive techniques such as postural drainage and percussion/vibration (commonly published in the 1970s, 1980s and early 1990s) to techniques such as PEP, oscillating PEP and autogenic drainage (commonly published in the mid to late 1990 s and 2000 s). There were two reasons for this. First, evidence emerged that in people with chronic lung disease, techniques such as postural drainage and percussion may lead to deleterious effects such as bronchospasm [97], desaturation [98] and gastro-oesophageal reflux [99]. Second, in contrast with percussion/vibration, the use of techniques such as PEP, oscillating PEP and autogenic drainage encourage people with CF to be more independent with their airway clearance, aligning with the principles of chronic disease self-management [93]. Despite being able to identify this trend in the type of ACT that were used, the exact nature of the ACT used were often described in insufficient detail that would allow replication. This reflects the fact that approximately 73% of the studies included in this review were published prior to 2014, when the importance of reporting intervention protocols and implementation fidelity was specifically articulated by the Template for Intervention Description and Replication (TIDieR) checklist [100].

Similar to earlier reviews of ACT in CF that have been published between 2015 and 2020 [16,17,91,92,93,94], there was limited capacity to complete a meta-analysis of between-group differences. The only previous review that completed a meta-analysis of studies in regard to sputum wet and dry weight pooled data across four studies and included data from two groups from within one crossover study (one RCT and three RXT). This earlier analysis reported no differences in wet or dry sputum weight when oscillating devices were compared to “conventional physiotherapy”, which comprised different groupings of postural drainage, percussion, vibration and clapping [16]. While the present study did not demonstrate that ACT produced more sputum when compared with cough alone, the mean summary effect statistic for all meta-analyses favoured the ACT group. Statistical heterogeneity was found in the meta-analyses of between-group changes in sputum wet weight when comparing PEP versus C and OT versus C. Due to this, a sensitivity analyses was run to reduce the statistical heterogeneity (I^2^ reduced to 0%), yet this did not change the finding of no between-group differences in sputum wet weight for the same comparisons. It is unlikely that robust RCT will become available to improve our precision around the estimate of this effect. This is because the strong physiological basis for these techniques, coupled with data from laboratory and animal studies [101,102], and enduring anecdotal evidence has resulted in acceptance by the clinical community that ACT are effective in people with CF. In fact, routine use of ACT in the management of adults with CF is recommended as a standard of care in national and international guidelines [6,103,104] and is likely to explain why most studies investigating ACT are RXT. Given the lack of data to support one ACT over all others, in clinical practice, ACT continue to be selected based on therapist and patient preference, the specific needs of the patient and accessibility of devices [16].

As future RCT of ACT in adults with CF are unlikely, a novel analysis was instead undertaken to explore possible moderators of within-group differences in the most commonly reported outcomes. The pooled estimate for sputum wet weight was 12.42 g [9.28 to 15.58]. The magnitude of this effect is difficult to interpret, as the minimal clinically important difference (MCID) for this outcome, in this population, has not been determined. Nevertheless, an important finding of this study was that compared with cough alone, greater sputum wet weight was reported by studies which applied any ACT. There are two important caveats to consider when interpreting this result. First, studies in this review were generally of low quality and at moderate-to-high risk of bias. Second, although sputum wet weight is a commonly used outcome that is easily interpretable by clinicians, its use as an outcome has been criticised. This is due to the estimate of wet weight sputum being potentially confounded by expectorated saliva and the use of muco-active medications (e.g., hypertonic saline). Nevertheless, these analyses suggest that ACT, when considered together, appear to offer additional benefit in terms of the amount of sputum expectorated, over and above cough alone. 

Finally, this study suggests that any immediate (i.e., less than 60 min) effect of ACT on FEV_1_ and RV is likely to be negligible. The MCID for FEV_1_ has been difficult to define in the CF population due to disease heterogeneity and age-related cofounders as a result of the growing lung; MCID values ranging from a 5–10% improvement have been proposed [105], which is greater than the change reported in this study. Our pooled estimate for change in FEV_1_ of 0.03 L falls short of the lower limit of the range of MCID for FEV_1_ of 0.10 L proposed for the management of other obstructive diseases including COPD [106]. This supports earlier work that has questioned the appropriateness of using FEV_1_ in studies evaluating ACT in adults with CF [105,107,108,109,110]. Despite these concerns, as mentioned previously FEV_1_ is the only measure of respiratory function that is recommended as an outcome by the European Medicines Agency [9] and the US Food and Drugs Administration [10] for clinical trials in people with CF. It appears that changes in FEV_1_ are not likely following ACT, irrespective of the type of ACT applied or the clinical stability of the study population. Similarly, the pooled estimate for change in RV of −0.14 L estimated in this study falls well below the MCID proposed for emphysema management of between −0.31 and −0.43 L [111]. One possible alternative outcome which has shown promise in measuring CF related lung disease and identifying treatment effects in patients with CF is lung clearance index measured via multiple breath washout (MBW) [105]; however, further evidence is needed before this outcome can be utilised as an adjunct or surrogate measure to FEV_1_. 

### Strengths, Limitations and Future Directions

Compared with earlier reviews in this area, the current study (i) extracted data across the largest number of studies (*n* = 68), (ii) described the variety in ACT and outcomes used, (iii) completed analyses of both between and within-group differences in sputum wet weight and FEV_1_ and (iv) completed meta-regression on within-group differences in sputum volume, dry weight and RV. A limitation of this study is the exclusion of studies published in any language other than English. Further, techniques such as postural drainage, percussion, autogenic drainage and external chest wall oscillation devices were grouped as OT; PEP and oscillatory PEP were grouped together. We accept that the mechanisms of action for these techniques are likely distinct, however, given the high risk of bias and serious inconsistency, indirectness and imprecision across the included studies, a pragmatic approach of grouping these techniques was chosen to ensure sufficient studies were included in the meta-regression. Although this choice of coding has not allowed us to look at the individual effects of ACT, it allowed us to look at ACT as a whole compared to cough alone. It is clear that future studies in this area need to report the intervention (and the implementation fidelity) according to the TIDieR guidelines [100].

## 5. Conclusions

The most common outcomes used to measure the effectiveness of ACT are not sensitive to the effects of ACT. More work is needed to explore alternative measures of respiratory function that can be used in RXT to compare the immediate effects of different ACT in adults with CF.

## Figures and Tables

**Figure 1 jcm-10-05280-f001:**
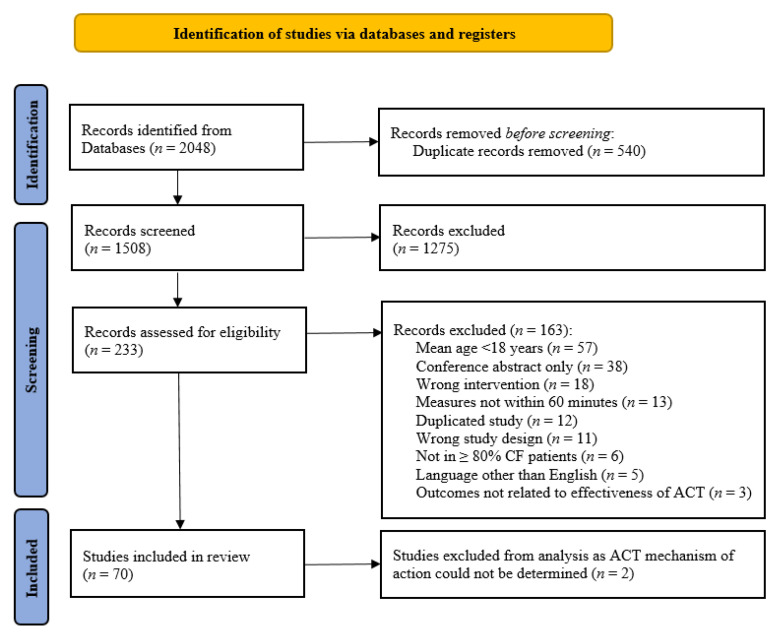
PRISMA study flow diagram [21].

**Figure 2 jcm-10-05280-f002:**
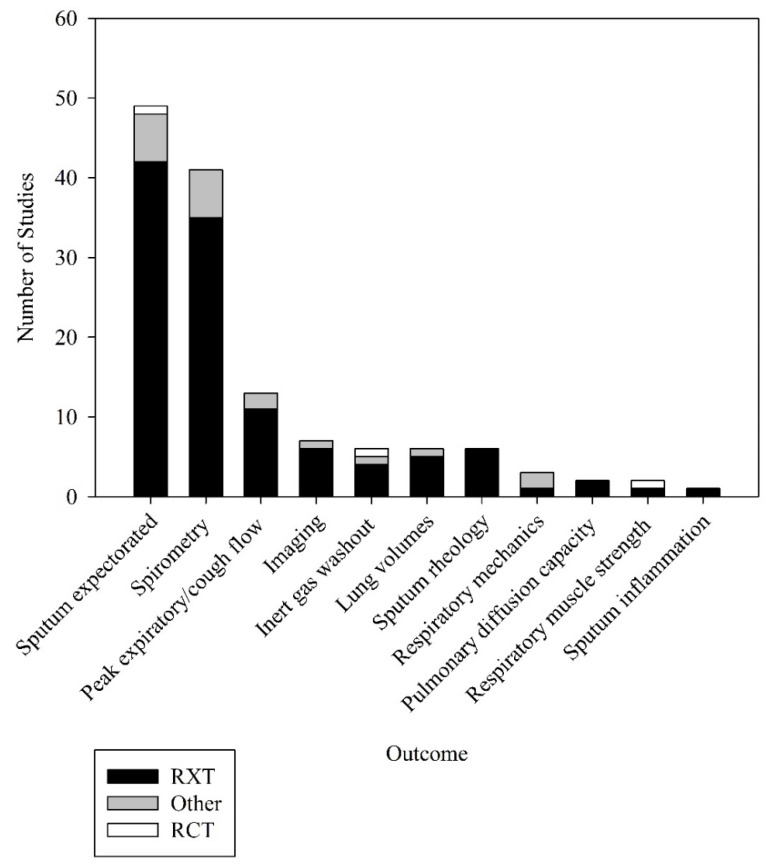
Objective outcomes used to measure effectiveness of airway clearance techniques. Other = other study designs apart from RXT and RCT; RCT = randomised controlled trials; RXT = randomised crossover trials.

**Figure 3 jcm-10-05280-f003:**
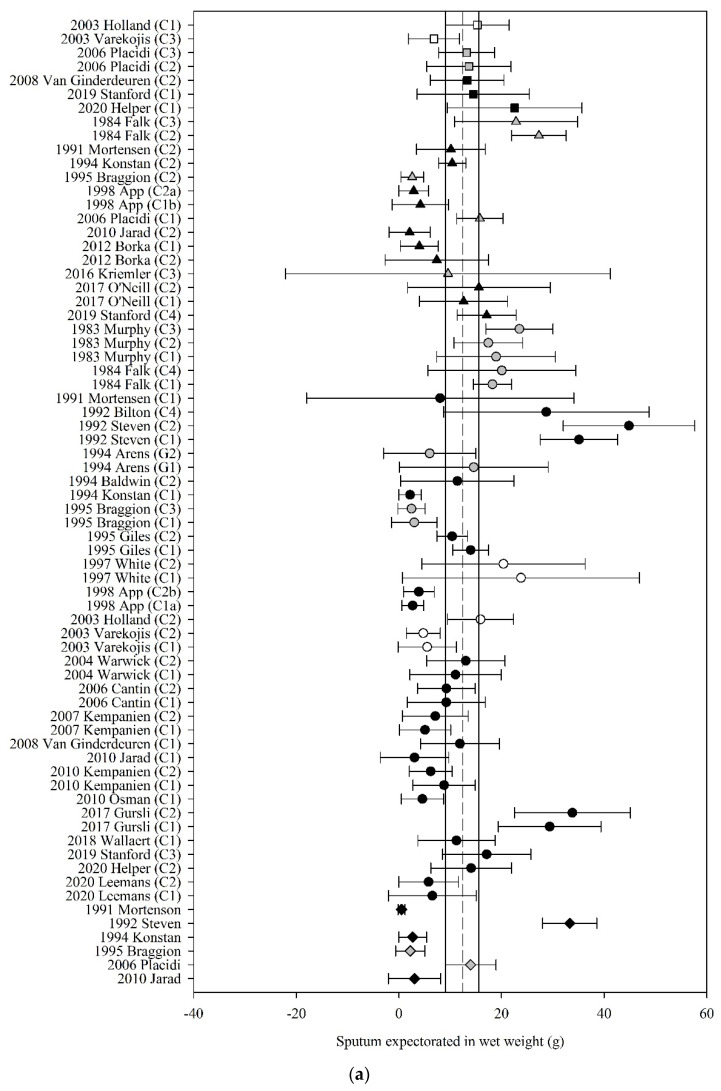
Pooled estimates–sputum expectoration: (**a**) sputum wet weight (grams); (**b**) sputum dry weight (grams); (**c**) sputum volume (millilitres). C_ = crossover intervention, G_ = RCT group, g = grams, ■ = positive pressure device (PPD), ▲ = positive expiratory pressure device (PEP), ● = other techniques (OT), ♦ = cough alone (C), closed symbol = stable population, open symbol = acute population, grey symbol = mixed population or nil info, dashed line = pooled estimate, solid vertical line = 95% CI, all individual data presented as mean ± SD.

**Figure 4 jcm-10-05280-f004:**
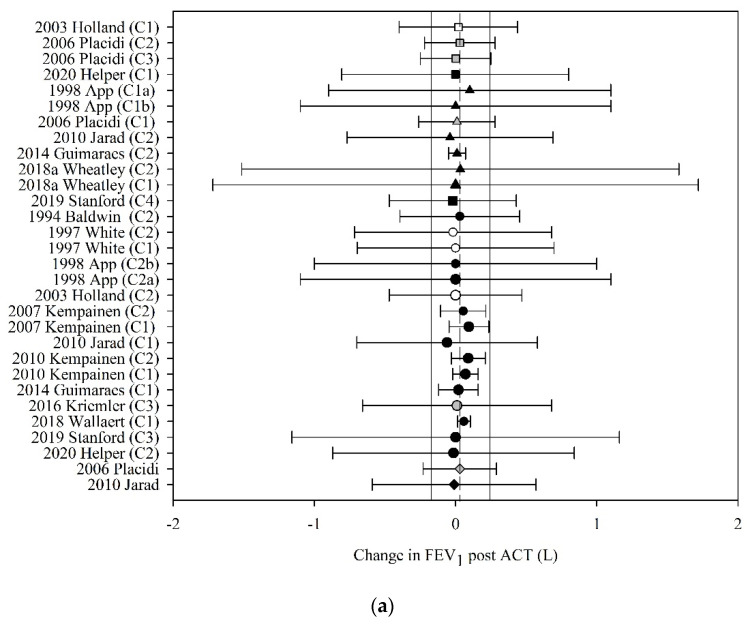
Pooled estimate-FEV1: (**a**) FEV1 (litres); (**b**) FEV1 (% predicted). ACT = airway clearance techniques, C_ = crossover intervention, L = litres, ■ = positive pressure device (PPD), % predicted = percent predicted, ▲ = positive expiratory pressure device (PEP), ● = other techniques (OT), ♦ = cough alone (C), closed symbol = stable population, open symbol = acute population, grey symbol = mixed population or nil info, dashed line = pooled estimate, solid vertical line = 95% CI, all individual data presented as mean ± SD.

**Figure 5 jcm-10-05280-f005:**
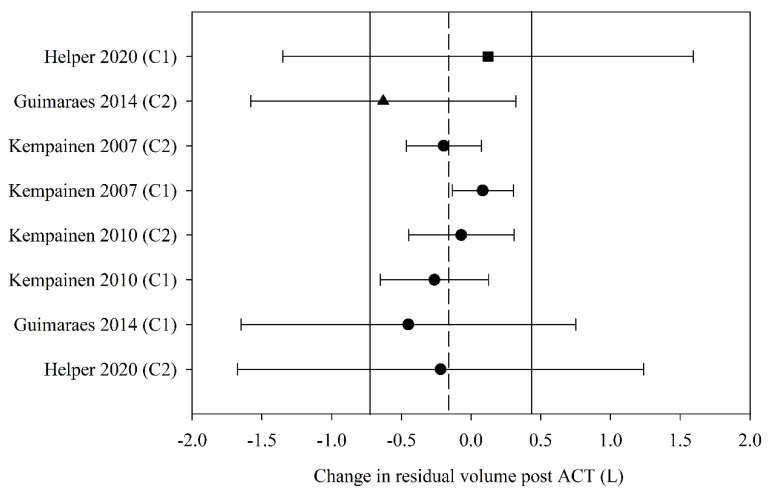
Pooled estimate–residual volume (litres). ACT = airway clearance techniques, C_ = crossover intervention, L = litres, ■ = positive pressure device (PPD), ▲ = positive expiratory pressure device (PEP), ● = other techniques (OT), closed symbol = stable population, dashed line = pooled estimate, solid vertical line = 95% CI, all individual data presented as mean ± SD.

**Table 1 jcm-10-05280-t001:** Summary of studies included in systematic review.

Study	Design	Sample	Clinical Status	Mechanism of Action and Airway Clearance Technique Used (G1, G2, C1, C2 Links with Figure 4 and Figure 5)	Outcomes and Useability (✓/✕/*) of These Data in Meta-Analyses
App 1998 [22]	RXT	*n* = 14 FEV_1_: 75%	Stable	C1: PEP (Flutter) [a = first, b = second] C2: OT (autogenic drainage)	Sputum wet weight ✓ Spirometry ✓
Aquino 2012 [23]	RXT	*n* = 15 FEV_1_: 54%	Stable	Cough alone C1: PPD (CPAP) C2: PPD (CPAP + hypertonic NaCl 7%)	Sputum volume ✓ Sputum rheology *
Arens 1994 [24]	RCT	*n* = 25 FEV_1_: 34% vs. 38%	Mix	G1: OT (external oscillation device) G2: OT (postural drainage & percussion)	Sputum wet and dry weight ✓
Baldwin 1994 [25]	RXT	*n* = 8 FEV_1_: 64%	Stable	C1: Exercise (aerobic)# C2: OT (postural drainage & ACBT & percussion & vibrations)	Sputum wet weight ✓ Spirometry ✓ Peak expiratory/cough flow *
Bilton 1992 [26]	RXT	*n* = 18 FEV_1_: N/A	Stable	C1: Exercise (cycling at 60%VO_2_max)# C2: Exercise (cycling at 60%VO_2_max then ACBT)# C3: Exercise (ACBT then cycling at 60%VO_2_max) # C4: OT (postural drainage & ACBT)	Sputum wet weight ✓ Spirometry ✕ Respiratory mechanics ✕
Bishop 2011 [27]	RXT	*n* = 17 FEV_1_: 67% vs. 68%	Stable	C1: Usual ACT (Dornase alpha after usual ACT)# C2: Usual ACT (Dornase alpha before usual ACT)#	Sputum wet weight ✕
Borka 2012 [28]	?RXT	*n* = 10 FEV_1_: 57%	Stable	C1: PEP (Flutter then PEP) C2: PEP (PEP then Flutter)	Sputum wet weight ✓
Braggion 1995 [29]	RXT	*n* = 16 FEV_1_: 53%	Mix	Cough alone C1: OT (external oscillation device) C2: PEP (PEP mask) C3: OT (postural drainage)	Sputum wet and dry weight ✓ Spirometry ✓
Cantin 2006 [30]	?RXT	*n* = 22 FEV_1_: 58%	Stable	C1: OT (external oscillation device) C2: OT (postural drainage & percussion)	Sputum wet weight ✓
Carr 1995 [31]	Other	*n* = 20 FEV_1_: 32%	Stable	OT (ACBT & percussion)	Sputum wet weight ✕
Chatham 2004 [32]	RXT	*n* = 20 FEV_1_: 60% vs. 48% (geometric mean)	Stable	C1: OT (resistive inspiratory manoeuvres) C2: OT (postural drainage & ACBT)	Sputum wet weight ✕ Sputum inflammatory markers ✕
Darbee 2004 [33]	RXT	*n* = 5 FEV_1_: 52%	Stable	Cough alone C1: PEP (PEP mask-low) C2: PEP (PEP mask-high)	Sputum wet and dry weight ✕ Spirometry ✕ Inert gas washout ✕ Lung volumes ✕
DeCesare 1982 [34]	Other	*n* = 10 FEV_1_: mild 68%, mod 43%, severe 24%	Mix	Cough alone C1: OT (postural drainage & percussion & vibrations)	Sputum volume ✕ Spirometry ✕ Peak expiratory/cough flow ✕ Imaging ✕
Dwyer 2011 [35]	RXT	*n* = 14 FEV_1_: 55%	Stable	Cough alone C1: Exercise (20min walk at 60%VO_2_max)# C2: Exercise (20min cycle at 60%VO_2_max)#	Sputum rheology * Peak expiratory/cough flow ✕
Dwyer 2015 [36]	RCT	*n* = 40 FEV_1_: 36% vs. 39%	Mix	G1: PPD (usual ACT & NIV) G2: Usual ACT#	Respiratory muscle strength *
Dwyer 2017 [37]	RXT	*n* = 25 FEV_1_: 51%	Stable	Cough alone C1: Exercise (20min walk at 60%VO_2_max)# C2: PEP (Flutter)	Sputum rheology * Peak expiratory/cough flow ✕
Dwyer 2019 [38]	RXT	*n* = 14 FEV_1_: 65%	Stable	Cough alone C1: Exercise (20min walk at 60%VO_2_max)# C2: PEP (Pari PEP)	Imaging *
Fainardi 2011 [39]	RXT	*n* = 34 FEV_1_: 67%	Acute	C1: OT (external oscillation device) C2: PEP (PEP mask)	Sputum volume ✓ Spirometry ✓
Falk 1984 [40]	RXT	*n* = 14 FEV_1_: 34%	Mix	C1: OT (postural drainage & percussion & huff/cough) C2: PEP (postural drainage & PEP mask & huff/cough) C3: PEP (PEP mask & huff/cough) C4: OT (pursed lip breathing & huff/cough)	Sputum wet weight ✓ Spirometry ✕ Peak expiratory/cough flow ✕
Giles 1995 [41]	RXT	*n* = 10 FEV_1_: N/A	Stable	C1: OT (postural drainage & percussion) C2: OT (autogenic drainage)	Sputum wet weight ✓ Spirometry ✓ Peak expiratory/cough flow *
Grosse-Onnebrink 2017 [42]	RCT	*n* = 41 FEV_1_: 57% vs. 51%	Acute	Cough alone G1: OT (external oscillation device)	Inert gas washout *
Guimaraes 2014 [43]	RXT	*n* = 14 FEV_1_: 34%	Stable	C1: OT (ELTGOL) C2: PEP (Flutter)	Sputum dry weight ✓ Spirometry ✓ Lung volumes ✓ Respiratory mechanics *
Gursli 2017 [44]	Other	*n* = 6 FEV_1_: 68%	Stable	C1: OT (huff) C2: OT (specific cough technique)	Sputum wet weight ✓ Spirometry ✓
Helper 2020 [45]	RXT	*n* = 22 FEV_1_: 54%	Stable	C1: PPD (MI-E Phillips E70 cough assist) C2: OT (autogenic drainage)	Sputum wet weight ✓ Spirometry ✓ Lung volumes ✓ Peak expiratory/cough flow *
Hofmeyr 1986 [46]	RXT	*n* = 18 FEV_1_: N/A	Stable	C1: PEP (PEP mask in sitting) C2: PEP (PEP mask & postural drainage) C3: OT (postural drainage & ACBT)	Sputum wet weight ✕ Spirometry ✕
Holland 2003 [47]	RXT	*n* = 26 FEV_1_: 34%	Acute	C1: PPD (ACBT & NIV) C2: OT (ACBT)	Sputum wet weight ✓ Spirometry ✓ Respiratory muscle strength *
Hordvik 1996 [48]	RXT	*n*= 24 FEV_1_: 56%	Acute	C1: OT (postural drainage & external percussive device & albuterol) C2: OT (postural drainage & external percussive device & 0.9% NaCl)	Spirometry *
Hortal 2014 [49]	Other	*n* = 16 FEV_1_: 56%	Stable	PEP (PEP & autogenic drainage & huff)	Spirometry ✓
Jarad 2010 [50]	RXT	*n* = 18 FEV_1_: N/A	Stable	Cough alone (sham hydro acoustic therapy) C1: OT (external oscillation device) C2: PEP (external oscillation device & Flutter)	Sputum wet and dry weight ✓ Spirometry ✓
Kempainen 2007 [51]	RXT	*n* = 15 FEV_1_: 72%	Stable	C1: OT (external oscillation device with sine waveform) C2: OT (external oscillation device with triangular waveform)	Sputum wet and dry weight ✓ Sputum rheology ✕ Spirometry ✓ Inert gas washout * Lung volumes ✓
Kempainen 2010 [52]	RXT	*n* = 16 FEV_1_: 68%	Stable	C1: OT (external oscillation device with higher pressure/variable frequency) C2: OT (external oscillation device with lower pressure/mid frequency)	Sputum wet and dry weight ✓ Sputum rheology ✕ Spirometry ✓ Inert gas washout * Lung volumes ✓
Konstan 1994 [53]	RXT	*n* = 17 FEV_1_: N/A	Stable	Cough alone C1: OT (postural drainage) C2: PEP (Flutter)	Sputum wet and dry weight ✓
Kriemler 2016 [54]	RXT	*n* = 12 FEV_1_: 63%	NA	C1: Exercise (trampolining then Flutter & thoracic expansion exercises & slow expiration)# C2: Exercise (cycling then Flutter & thoracic expansion exercises & slow expiration)# C3: PEP (billiards then Flutter & thoracic expansion exercises & slow expiration)	Sputum wet weight ✓ Spirometry ✓
Lannefors 1992 [55]	RXT	*n* = 9 FEV_1_: 51%	Stable	C1: PEP (PEP mask & huff) C2: Exercise (cycling & huff)# C3: OT (postural drainage & thoracic expansion exercises & huff)	Imaging ✕
Leemans 2020 [56]	RXT	*n* = 8 FEV_1_: 61% vs. 62%	Stable	C1: OT (mobile external oscillation device) C2: OT (external oscillation device)	Sputum wet weight ✓ Sputum volume ✓
Lyons 1993 [57]	RXT	*n* = 12 FEV_1_: N/A	Stable	C1: PEP (Flutter) C2: PEP (usual ACT & Flutter) C3: OT (usual ACT & sham Flutter)# C4: Usual ACT#	Peak expiratory/cough flow ✕
McCarren 2006 [58]	RXT	*n* = 18 FEV_1_: 55%	Stable	Cough alone C1: OT (vibrations) C2: OT (percussion) C3: PEP (PEP mask) C4: PEP (Flutter) C5: PEP (Acapella) C6: OT (total lung capacity with passive expiration) C7: OT (high lung volume huff)	Peak expiratory/cough flow ✕
Mentore 2005 [59]	Other	*n* = 8, FEV_1_: 67%	N/A	Chest PT – nil other info#	Imaging ✕
Milne 2004 [60]	RXT	*n* = 7 FEV_1_: NA	N/A	C1: PEP (Flutter) C2: OT (ACBT)	Spirometry ✕ Peak expiratory/cough flow ✕
Mortensen 1991 [61]	RXT	*n* = 10 FEV_1_: 47%	Stable	Cough alone C1: OT (postural drainage & huff) C2: PEP (PEP mask & huff)	Sputum wet weight ✓ Imaging *
Murphy 1983 [62]	RXT	*n* = 2FEV_1_: NA	N/A	C1: OT (postural drainage & external percussive device) C2: OT (postural drainage & percussion) C3: OT (postural drainage & huff)	Sputum wet weight ✓ Spirometry ✕ Peak expiratory/cough flow ✕
O’Neill 2017 [63]	RXT	*n* = 13 FEV_1_: 51%	Stable	C1: PEP (hypertonic NaCl pre Acapella Duet) C2: PEP (hypertonic NaCl during Acapella Duet)	Sputum wet weight ✓ Inert gas washout *
Osman 2010 [64]	RXT	*n* = 29 FEV_1_: 38%	Stable	C1: OT (external oscillation device) C2: Usual ACT#	Sputum wet weight ✓ Spirometry ✓
Pfleger 2015 [65]	Other	*n* = 29 FEV_1_: 70%	Stable	PEP (PEP mask)	Spirometry ✕ Inert gas washout ✕ Lung volumes ✕ Respiratory mechanics ✕
Placidi 2006 [66]	RXT	*n* = 17 FEV_1_: 25%	Mix	Cough alone C1: PEP (PEP mask) C2: PPD (CPAP) C3: PPD (NIV)	Sputum wet and dry weight ✓ Spirometry ✓
Pryor 1979a [67]	RXT	*n* = 16 FEV_1_: N/A	Stable	C1: OT (postural drainage & thoracic expansion exercises & shakes & therapist percussion & cough) C2: OT (postural drainage & thoracic expansion exercises & self-percussion & huff & cough)	Spirometry ✕ Peak expiratory/cough flow ✕
Pryor 1979b [67]	RXT	*n* = 8 FEV_1_: N/A	Mix	C1: OT (postural drainage & thoracic expansion exercises & self-percussion & huff & cough) C2: OT (postural drainage & shakes & therapist percussion & huff)	Spirometry ✕ Peak expiratory/cough flow ✕
Pryor 1979c [68]	RXT	*n* = 24 FEV_1_: N/A	Stable	C1: OT (postural drainage & thoracic expansion exercises & percussion & cough) C2: OT (postural drainage & thoracic expansion exercises & percussion & huff & cough)	Sputum wet weight ✕ Spirometry ✕
Pryor 1981 [69]	RXT	*n* = 12 FEV_1_: NA	Stable	C1: OT (postural drainage & external percussive device & huff) C2: OT (postural drainage & percussion & huff)	Sputum wet weight ✕ Spirometry ✕
Pryor 1990 [70]	Other	*n* = 20 FEV_1_: N/A	N/A	OT (postural drainage & ACBT)	Sputum wet weight ✕
Pryor 1994 [71]	RXT	*n* = 20 FEV_1_: N/A	Stable	C1: OT (ACBT) C2: PEP (Flutter & ACBT)	Sputum wet weight ✕ Spirometry ✕
Radtke 2018 [72]	RXT	*n* = 15 FEV_1_: 56%	Stable	C1: PEP (Flutter & interval cycling) C2: Exercise (continuous cycling)#	Sputum rheology * Pulmonary diffusion capacity *
Robinson 1996 [73]	RXT	*n* = 12 FEV_1_: 61%	Stable	Cough alone C1: 0.9% NaCl# C2: Amiloride in 0.12% NaCl# C3: Hypertonic NaCl 7%# C4: Amiloride & hypertonic NaCl 7%#	Spirometry ✓ Imaging ✕
Robinson 1997 [74]	RXT	*n* = 10 FEV_1_: 52%	Stable	Cough & 0.9% NaCl C2: Hypertonic NaCl 3%# C3: Hypertonic NaCl 7%# C4: Hypertonic NaCl 12%#	Spirometry ✓ Imaging ✕
Rossman 1982 [75]	RXT	*n* = 6 FEV_1_: 38%	Stable	Cough alone (spontaneous) Cough alone 2 (directed) C1: OT (postural drainage) C2: OT (postural drainage & external percussive device) C3: OT (postural drainage & thoracic expansion exercises & vibrations & therapist percussion)	Sputum volume ✕ Imaging *
San Miguel-Pagola 2020 [76]	RXT	*n* = 22 FEV_1_: 67%	Stable	C1: PEP (Hypertonic NaCl 7% & hyaluronic acid 0.1% via Acapella Duet pre autogenic drainage) C2: OT (Hypertonic NaCl 7% & hyaluronic acid 0.1% neb pre autogenic drainage)	Sputum volume ✓
Scherer 1998 [77]	RXT	*n* = 14 FEV_1_: 59%	Stable	C1: PPD (Sensormedics Oscillatory oral Ventilator 3100B) C2: PPD (Sensormedics MCT1 oral airway oscillator) C3: OT (external oscillation device at optimal settings) C4: OT (external oscillation device using secretion clearance mode) C5: OT (postural drainage & percussion & external percussive device & cough)	Percentage of baseline wet and dry sputum expectorated * Spirometry *
Sokol 2015 [78]	Other	*n* = 40 vs. 32 FEV_1_: 57% vs. 66%	Stable	G1: PEP (Tri-Gym incentive spirometer) G2: OT (autogenic drainage)	Spirometry * Peak expiratory/cough flow *
Stanford 2019 [79]	RXT	*n* = 14 FEV_1_: 49%	Stable	C1: PPD (NIV & usual ACT) C2: PEP (PEP/oscillating PEP) C3: OT (ACBT & autogenic drainage)	Sputum wet weight ✓ Spirometry ✓
Steven 1992 [80]	RXT	*n* = 24 FEV_1_: N/A	Stable	Cough alone C1: OT (ACBT & cough in sitting) C2: OT (postural drainage & ACBT & cough)	Sputum wet weight ✓ Spirometry ✕
Van Ginderdeuren 2008 [81]	RXT	*n* = 20 FEV_1_: 65%	Stable	C1: OT (0.9% NaCl neb pre autogenic drainage) C2: PPD (0.9% NaCl via IPV pre autogenic drainage)	Sputum wet weight ✓
Varekojis 2003 [82]	RXT	*n* = 24 FEV_1_: 55%	Acute	C1: OT (postural drainage & percussion) C2: OT (external oscillation device) C3: PPD (IPV & 0.9% NaCl)	Sputum wet and dry weight ✓
Verboon 1986 [83]	RXT	*n* = 8 FEV_1_: 38%	Stable	C1: OT (huff post sleeping in postural drainage position) C2: OT (huff in upright post sleeping in horizontal position)	Sputum wet weight ✕ Spirometry ✕ Peak expiratory/cough flow ✕
Wallaert 2018 [84]	Other	*n* = 30 FEV_1_: 40%	Stable	OT (autogenic drainage)	Sputum wet and dry weight ✓ Spirometry ✓ Respiratory mechanics *
Warwick 2004 [85]	RXT	*n* = 12 FEV_1_: N/A	Stable	C1: OT (percussion & huff & directed coughing) C2: OT (external oscillation device & huff)	Sputum wet and dry weight ✓
Webber 1985 [86]	RXT	*n* = 16 FEV_1_: 32%	Stable	C1: OT (postural drainage & thoracic expansion exercises & percussion & huff) C2: OT (postural drainage & thoracic expansion exercises & huff)	Sputum wet weight✕ Spirometry ✕
Webber 1986 [87]	Other	*n* = 11 FEV_1_: 61%	Stable	OT (postural drainage & ACBT & percussion & vibrations & shakes)	Sputum wet weight ✕
Wheatley 2018a [88]	RXT	*n* = 10 FEV_1_: 76%	Stable	C1: PEP (Vibralung sham) C2: PEP (Vibralung)	Spirometry ✓ Pulmonary diffusion capacity *
Wheatley 2018b [88]	RXT	*n* = 11 FEV_1_: 57%	Acute	C1: PEP (Vibralung day 1–5) C2: PEP (Vibralung day 7–11) C3: OT (external oscillation device)	Sputum wet and dry weight ✕
White 1997 [89]	RXT	*n* = 15 FEV_1_: mild (>70%) = 3, mod (41–70%) = 4, severe (≤40) = 8	Acute	C1: OT (postural drainage & ACBT) C2: OT (postural drainage & ACBT without thoracic expansion exercises)	Sputum wet weight ✓ Spirometry ✓ Peak expiratory/cough flow ✕

All FEV_1_ data are means (conversion from median to mean occurred as necessary [90]). External oscillation device includes devices that were used to exert oscillations onto the chest wall externally such as the Vest^®^ (Hill-Rom, Batesville, Indiana) and the Monarch® Airway Clearance System (Hill-Rom, Batesville, Indiana). External percussive device includes devices that were used to provide percussion to the chest wall externally such as the EquiMed Percussor (Medical and Scientific Computer Services Ltd, Lisburn, N. Ireland). Footnote: ACBT = active cycle of breathing techniques, ACT = airway clearance techniques, C = crossover group related to RXT, CPAP = continuous positive airway pressure, ELTGOL = L’Expiration lente total glotte ouverte en de-cubitus lateral, G = group related to RCT, IPV = intrapulmonary percussive ventilation, N/A = data not available, NIV = non-invasive ventilation, OT= other techniques, Other = study design other than RCT/RXT, PEP = positive expiratory pressure, PPD = positive pressure device, RCT = randomised controlled trial, RXT = randomised crossover trial, ?RXT = crossover trial with randomisation not specified, # = groups removed from analyses as did not meet study criteria for ACT or could not be coded by mechanism of action, ✓ = data used in meta-analyses, ✕ = insufficient data to be included in meta-analyses, *** = sufficient data for meta-analyses however unable to complete due to <four studies.

## Data Availability

Data are available from corresponding author upon request.

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
