# Peer review of "Methods Used to Evaluate the Immediate Effects of Airway Clearance Techniques in Adults with Cystic Fibrosis: A Systematic Review and Meta-Analysis"

_jcm, 2021, doi:10.3390/jcm10225280_

Round 1
Reviewer 1 Report
Thank you for the opportunity to review this manuscript. The paper addresses an area of clinical uncertainty and the differences in the methods used to evaluate airway clearance techniques in adults with cystic fibrosis. The topic of the paper is potentially interesting to international readers, and it is well-written. I have only one minor comment.
Figure 1: Please check the numbers are correct. After excluding 162 records from those assessed for eligibility (n = 235), the number of remaining records would be 73. However, the number is 72 in Figure 1, comprising studies included in the review (n = 70) and those excluded (n = 2).
Author Response
Response to reviewer comments for Manuscript ID: jcm-1419105 titled “Methods used to evaluate the immediate effects of airway clearance techniques in adults with cystic fibrosis: a systematic review and meta-analysis.”
We would like to thank you for your time and valuable comment. We believe that the manuscript has definitely improved as a result of this suggestion. Please find below a response to your comment. All changes made to the manuscript in response to these comments can be identified using tracked changes.
REVIEWER #1
Comment 1: Figure 1: Please check the numbers are correct. After excluding 162 records from those assessed for eligibility (n = 235), the number of remaining records would be 73. However, the number is 72 in Figure 1, comprising studies included in the review (n = 70) and those excluded (n = 2).
Response 1: Thank you for noticing this inconsistency. We reviewed the numbers in Figure 1 and made the appropriate changes in the main manuscript document, please see the updated Figure 1, in section 3.1 (page 4).
Reviewer 2 Report
I thank the editor for the invitation to review this manuscript. In general, it seems that it is a good summary of the available literature and that the meta-analysis strengthens the results. Therefore, I only have a few comments that the authors must respond to before recommending possible acceptance.
Major comments
- The title of the manuscript and its objective include the evaluation methods. However, the introduction hardly covers these methods. The authors describe most are the techniques, manual or instrumental, which are ultimately the interventions, not the evaluation methods. Please add information regarding the different methods in the introduction.
- Please check the sums in the flow chart. 1508 articles were screened, and 1275 were excluded, leaving 233 (not 235). After 235, 162 were excluded, leaving 73 to be included (not 70). Please revise.
- There are enough RCTs and RXTs. So why did the authors decide to include other lower-quality designs?
- Please, include in the figures of meta-analysis the final pooled estimation of each variable
- Interestingly, the D3 and D4 domains of risk of bias had all studies at low risk of bias (RXT) . Could the authors summarize the result according to the domain in the section results / risk of bias?
- In the supplementary material S2 a) and b), there is heterogeneity close to 90%. The authors should add this point to the discussion. While they name the heterogeneity of the disease, they do not name the statistical heterogeneity.
- Please add the PRISMA checklist as supplementary material.
Author Response
Response to reviewer comments for Manuscript ID: jcm-1419105 titled “Methods used to evaluate the immediate effects of airway clearance techniques in adults with cystic fibrosis: a systematic review and meta-analysis.”
We would like to thank you for your time and valuable comments. We believe that the manuscript has definitely improved as a result of these suggestions. Please find below a point-by-point response to the comments and suggestions. All changes made to the manuscript in response to these comments can be identified using tracked changes.
REVIEWER #2
Comment 1: The title of the manuscript and its objective include the evaluation methods. However, the introduction hardly covers these methods. The authors describe most are the techniques, manual or instrumental, which are ultimately the interventions, not the evaluation methods. Please add information regarding the different methods in the introduction.
Response 1: Thank you for your comment. We have included further information in the introduction regarding the measures used to evaluate the immediate effects of ACT. The new sentences read as follows: “Several objective outcomes have been used to evaluate the immediate effects of ACT as well as the medium- and long- term effects of these techniques. However, there is little agreement about which are to be used at each time interval post ACT. The primary outcome for respiratory function recommended by the European Medicines Agency [9] and the US Food and Drugs Administration [10] for clinical trials in people with CF is spirometry; in particular, forced expiratory volume in one second (FEV1). Other examples of objective outcomes used to evaluate the immediate effects of ACT include spu-tum expectoration (as either wet/dry weight or volume), inflammatory markers and rheology, measurements of lung volumes, lung mechanics, diffusion capacity and various imaging modalities.” Please see section 1 (page 2; lines 64 to 73) of the main manuscript for this addition.
Comment 2: Please check the sums in the flow chart. 1508 articles were screened, and 1275 were excluded, leaving 233 (not 235). After 235, 162 were excluded, leaving 73 to be included (not 70). Please revise.
Response 2: Thank you for noticing this inconsistency. We reviewed the numbers in Figure 1 and made the appropriate changes in the main manuscript document, please see the updated Figure 1, in section 3.1 (page 4).
Comment 3: There are enough RCTs and RXTs. So why did the authors decide to include other lower-quality designs?
Response 3: Thank you for your query. Prior to undertaking this review we were uncertain about the number of RCT and RXT that would meet our study criteria, therefore we decided to include all study designs. We agree that it is important to recognise that RCT and RXT are higher quality which is why we have chosen to state the study design in table 1.
Comment 4: Please, include in the figures of meta-analysis the final pooled estimation of each variable
Response 4: Thank you for this suggestion. We have now included these in each figure for all variables. Please see manuscript document for changes to Figure 3a, b and c (page 14 & 15), Figure 4a and b (page 16 & 17), along with Figure 5 (page 18).
Comment 5: Interestingly, the D3 and D4 domains of risk of bias had all studies at low risk of bias (RXT). Could the authors summarize the result according to the domain in the section results / risk of bias?
Response 5: Thank you for your suggestion. As you mentioned in the RXT, in D3 (bias due to missing outcome data) all except for one study were rated as low risk of bias and in D4 (measurement of the outcome) all studies were rated as low risk of bias. Please see section 3.1 (page 13; lines 191 to 196) of the main manuscript for amendments. The new sentences read as follows: “All RXT were judged as having a low risk of bias in regards to measurement of the outcome and all but one RXT were judged as having low risk of bias due to missing outcome data. The most common domains for high or unclear risk of bias in RXT were related to bias arising in the selection of the reported results (n=49 studies; 86%), from the randomisation process (n=39 studies; 68%) and from deviations from intended interventions (n=28 studies; 59%).”
Comment 6: In the supplementary material S2 a) and b), there is heterogeneity close to 90%. The authors should add this point to the discussion. While they name the heterogeneity of the disease, they do not name the statistical heterogeneity.
Response 6: Thank you for this suggestion. We have added this point to the discussion of the main manuscript in section 4 (page 19; line 339 to 343). The new sentences read as follows “Statistical heterogeneity was found in the meta-analyses of between-group changes in sputum wet weight when comparing PEP versus C and OT versus C. Due to this, a sensitivity analyses was run to reduce the statistical heterogeneity (I2 reduced to 0%), yet this did not change the finding of no between-group differences in sputum wet weight for the same comparisons.”
Comment 7: Please add the PRISMA checklist as supplementary material.
Response 7: Please see updated supplementary material file for completed PRISMA checklist.